

# An adaptive weighting mechanism for Reynolds rules-based flocking control scheme

Duc N. M. Hoang[1,2], Duc M. Tran[1,2], Thanh-Sang Tran[1,2] and Hoang-Anh Pham[1,2]

[1] Faculty of Computer Science and Engineering, Ho Chi Minh City University of Technology (HCMUT), Ho Chi Minh, Vietnam
[2] Vietnam National University of Ho Chi Minh City (VNU-HCM), Ho Chi Minh, Vietnam

## ABSTRACT

Cooperative navigation for fleets of robots conventionally adopts algorithms based on Reynolds's flocking rules, which usually use a weighted sum of vectors for calculating the velocity from behavioral velocity vectors with corresponding fixed weights. Although optimal values of the weighting coefficients giving good performance can be found through many experiments for each particular scenario, the overall performance could not be guaranteed due to unexpected conditions not covered in experiments. This paper proposes a novel control scheme for a swarm of Unmanned Aerial Vehicles (UAVs) that also employs the original Reynolds rules but adopts an adaptive weight allocation mechanism based on the current context than being fixed at the beginning. The simulation results show that our proposed scheme has better performance than the conventional Reynolds-based ones in terms of the flock compactness and the reduction in the number of crashed swarm members due to collisions. The analytical results of behavioral rules' impact also validate the proposed weighting mechanism's effectiveness leading to improved performance.

## INTRODUCTION

Swarms of UAVs have recently achieved growing popularity in both industrial and research fields due to the development of UAV technology and reasonably priced models, namely drones or quadcopters (*Bürkle, Segor & Kollmann, 2011*). Compared with a single quadcopter, maneuvering a large group of these vehicles has many advantages (*Seeja, Arockia Selvakumar & Berlin Hency, 2018*; *Tahir et al., 2019*). For instance, fireworks are being replaced with light displays where manifold LED-equipped drones are controlled by an algorithm recreating objects from their flight patterns (*Hörtner et al., 2012*). Besides, UAV swarms can take soldiers' place to sweep a large area in aerial reconnaissance missions (*Wang et al., 2019*). Since they are typically flown uninhabited, flocks of remotely piloted aircraft can be utilized in lethal situations such as gas leakage detection (*Braga et al., 2017*) or firefighting (*Innocente & Grasso, 2019*).

Corresponding author
Hoang-Anh Pham,
anhpham@hcmut.edu.vn

Although there are many benefits, multi-UAV systems still face certain challenges. *Shakeri et al. (2019)* demonstrated several architectures and design issues, including camera coverage of areas and objectives, control strategies for trajectory planning, image processing and vision-based methods, communication aspects, and low-level flight control. In practice, building a path planning scheme is one of the most fundamental yet crucial tasks as the basic activity of any swarms is to travel in groups before performing any particular tasks collectively. Besides, it requires that there is no collision not only among swarm members but also between an individual and its surroundings. Moreover, as wireless communication is usually deemed the backbone of a multi-UAV system, the problem arises when the aircraft have to keep a safe distance while maintaining adequate signal coverage among one another (*Brust et al., 2012*; *Dai et al., 2019*).

Possible solutions to swarm control methods are to adopt well-known clustering schemes. For example, *Bannur et al. (2020)* proposed a swarm control system responsible for obstacle avoidance and path planning based on Cohort Intelligence methodology (*Kulkarni, Durugkar & Kumar, 2013*), but refined for applications in dynamic environments. For UAV swarms using Flying ad-hoc Network (FANET), *Khan, Aftab & Zhang (2019)* implemented a self-organized flocking scheme by a behavioral study of Glowworm Swarm Optimization (GSO). The proposed system offered a wide range of functions such as route planning, cluster leader election, and cluster formation.

In addition, graph theory or topology can be studied to develop control schemes for multi-agent coordination systems. *Olfati-Saber (2006)* introduced a graph-theory-based framework to design and analyze three proposed algorithms for both distributed obstacle-free and constrained flocking. *Shucker, Murphey & Bennett (2008)* presented a cooperative control system that ensures the stability for network graphs, even in the graph topology switching. Likewise, *Ning et al. (2018)* investigated the *dispersion* and *flocking* behaviors by employing the acute angle test (AAT)-based rule for interactions with distant neighbors. Thanks to switching topology, this work has proved scalable, robust, and able to tackle the aforementioned signal coverage problem.

Other feasible and worth noting solutions are those based on the Reynolds rules (*Reynolds, 1987*). In such flocking schemes, each swarm member can perceive the surrounding environment, then makes its own movement decisions. A typical Reynolds-based flocking control scheme usually follows an iterative procedure adopted by every member in the entire swarm of $N$ quadcopters to calculate the velocity vector and update its position. For example, quadcopter $i$th ($i \in \{1,2,\ldots,N\}$) will determine its new position $y_i$ by $y_i = x_i + vt$ where $x_i$ is the current position vector, $v$ denotes the velocity vector affected by a set of swarm rules $R$, and $t$ is the time traveled. Moreover, $v$ is calculated by (1).

$$v = u_i + \sum_{r=1}^{|R|} w_r z_r \qquad (1)$$

where $u_i$ is the current velocity vector, $w_r$ and $z_r$ are the weight and the velocity vector for rule $r$th, respectively. That is to say, when cooperatively traveling in groups, quadcopters are directed by steering their current heading to the desired one, which is

represented by the sum of all vectors $z_r$ generated by flocking behaviors. The act of calculating $z_r$ is also the feature of self-organizing flight schemes.

Conventionally, each rule's weight $w$ is fixed throughout the flight. The higher it is, the more its corresponding swarm rule affects the quadcopter's behaviors. However, a problem arises as to how to determine an optimal set of rule weights, and thus it is essential to employ an adaptive weighting mechanism for the following reasons. Firstly, a set of rule weights not carefully chosen may lead to a considerable number of collisions between quadcopters due to excessive steering forces from swarm behaviors. Besides, optimal weights can usually be determined only after conducting a large number of experiments, which is time-consuming in general. Moreover, an adaptive weighting system will enable swarm members to be more robust against in-flight changes (e.g., an increasing number of swarm members).

However, developing an adaptive weighting scheme faces certain challenges. The scheme itself should adapt to different environments regardless of the length unit used in their coordinate system. Additionally, an effective weighting system should deal with the swarm's common problems, such as reducing the swarm size or collision avoidance between UAVs. In this study, we propose an adaptive mechanism that allocates weights for each swarm rule's velocity vector in every new step based on the current context rather than being fixed from the beginning. Meanwhile, our proposed method can also reduce the number of quadcopter crashes, compact the swarm shape, and work under various environment settings.

## RELATED WORK

Over the years, many studies have been conducted to address the cooperative movement of aerial robots. *Reynolds (1987)* developed an algorithm that simulates flocking behaviors of flocks of birds. In that work, simulated entities utilize position and velocity information of adjacent neighbors and their locomotion to make their next movements cooperatively based on specific behaviors. This flocking model's simplest form comprises three fundamental behavioral rules, including separation, alignment, and cohesion.

- *Separation* means each individual tends to avoid collisions with nearby flock mates by steering away from them.
- *Alignment* means each individual tends to match their own velocity with the average velocity of nearby flock mates.
- *Cohesion* means each individual tends to move to the average position of its nearby flock mates.

In addition to the three above basic rules, more functional and complex swarm models in robotics use other ones as well (*Schranz et al., 2020*). *Braga et al. (2018)* incorporated *Migration* rule to drive a drone swarm to destinations while swarm members perform collision avoidance with each other. However, the implementation is relatively simple because the weight coefficients are fixed. Additionally, for each particular scenario, the weight coefficients' values can only be found manually to achieve the best performance.

Other uses of Reynolds rules also include flocking navigation for drone groups in search of gas emitting sources (*Braga et al., 2017*). Although the Particle Swarm Optimization algorithm (*Kennedy & Eberhart, 1995*) is used to compute the velocity vector for one behavior, the values of other traditional rule weights still require to be empirically acquired.

*Watson, John & Crowther (2003)* implemented a graphical simulation of UAV flocking with Reynolds rules. The program allows users to interactively tune weightings of swarm rules to explore practical applications of commanding UAVs with flocking algorithms. *Sun & Tokunaga (2014)* integrated external dynamic conditions into the traditional Reynolds rules for more realistic flocking. Despite being able to be adjusted during the simulation process, it is time-consuming to obtain optimal rule weights and environment parameters.

*Clark & Jacques (2012)* conducted UAV flight tests with the *waypoint-following* control system that uses the Boids Guidance Algorithm optimized by a Simple Genetic Algorithm. They introduced various combinations of behavior weightings, but due to the requirement of high-performance computing resources, only one set of behaviors were selected for optimization. The remaining ones were set manually so the overall performance could be affected by unexpected conditions in some particular scenarios.

*Kownacki & Ołdziej (2015)* were also inspired by the Reynolds rules and then introduced a flight control scheme for fixed-wing vehicles that cannot change their orientation in a place. Due to the leadership mechanism's application, only the *Cohesion* rule was globally implemented among the entire swarm members. Although the local *Separation* rule was partly online weighted based on the actual distance during flights, the system's hierarchical structure failed to cope with the sole leader's malfunction as there was no leader re-election system.

Recently, *Huang, Tang & Lao (2019)* carried out an analysis of the collision avoidance method for fixed-wing UAVs. The authors proposed a flocking optimization algorithm based on a fitness function to evaluate the flight effect. However, the analysis mainly focused on the anti-collision force's effect but not the effect of weight coefficients upon the simulation scenarios. Only one set of weight coefficients is used to analyze the results while generating the set is not indicated clearly.

## THE PROPOSED APPROACH

Besides three traditional Reynolds rules, we also employ the *Migration* rule like *Braga et al. (2018)* to drive the UAV swarm to a specific location. In addition, the *Avoidance* rule is proposed and added to navigate the entire swarm away from stationary obstacles. Therefore, there are totally five behavioral rules adopted in our proposed control scheme, including *Alignment*, *Avoidance*, *Cohesion*, *Migration*, and *Separation*. As we examine the use of Reynolds rules for UAV swarm applications, it is required to consider which behaviors should be more particularly important based on real-world concerns. Therefore, we sort the swarm behaviors in priority order as partially suggested by *Allen (2018)*, and then apply a novel adaptive weighting mechanism. However, in our proposed approach, the behaviors are grouped by their similarity in contextual significance

instead of being separately prioritized. Hence, those behaviors in a group will share the same weighting function and the same value range of weight, ensuring that less important behaviors will not supersede those of greater importance.

### Behavioral rule prioritization

The five above behavioral rules are divided into three groups of behaviors according to their contextual significance.

- Both *Separation* and *Avoidance* should be given the highest priorities due to safety issues. In cases where they are both obstacles and neighbors violating the safety distance, the drone will attempt to avoid the nearest object.
- The *Migration* rule is the second most crucial behavior since its role is to navigate the group of drones to destinations as planned.
- The *Alignment* and *Cohesion* rules are used to make the drones travel in a swarm and maintain the compactness. They are the lowest priorities.

### Adaptive weighting mechanism

The weights of the five behavioral rules, including $w_{se}$ (*Separation*), $w_{av}$ (*Avoidance*), $w_{mi}$ (*Migration*), $w_{al}$ (*Alignment*), and $w_{co}$ (*Cohesion*) are determined by (2)–(6), respectively.

$$w_{se} = T_s \max_{i \in I, i \neq k} \left( c_s - \frac{d_{i,k}^2}{c_s} \right) \forall k \in m \tag{2}$$

$$w_{av} = T_a \max_{i \in I, i \neq k} \left( c_a - \frac{d_{i,k}^2}{c_a} \right) \forall k \in m \tag{3}$$

$$w_{mi} = \begin{cases} M, & \text{if } m = 0 \\ M + \dfrac{M}{m}, & \text{if } m \neq 0 \end{cases} \tag{4}$$

$$w_{al} = \begin{cases} 0, & \text{if } m = 0 \\ \dfrac{M}{m}, & \text{if } m \neq 0 \end{cases} \tag{5}$$

$$w_{co} = \begin{cases} 0, & \text{if } m = 0 \\ \dfrac{M}{m}, & \text{if } m \neq 0 \end{cases} \tag{6}$$

where

- $m$ denotes the number of nearby objects at a certain moment. An object within a quadcopter's *perception* will be considered as its nearby neighbor in *Separation*, *Cohesion*, *Alignment*, and *Migration* rules or obstacles in *Avoidance* rule.
- $c_s$ and $c_a$ are perception coefficients for the *Separation* and *Avoidance* rules, respectively.

- $d_{j,k}$ is the distance between quadcopters $i$th and $k$th (*in Separation*); or between quadcopter $i$th and obstacle $k$th (*in Avoidance*), $k \in \{1,2,\dots,m\}$.
- $M$ is the optimal number of neighbors surrounding each quadcopter. In an ideal case, each of the six quadcopters will try to maintain the same relative distance with the one at the center and with two adjacent ones, consequently forming a hexagon with quadcopters at its center and vertices, as illustrated in Fig. 1. Therefore, for a swarm with a large enough number of members as in our implementation, $M$ will be six (i.e., $M = 6$).
- $T_s$ and $T_a$ are the *transformation factors* for *Separation* and *Avoidance* rules, respectively.

## Transformation factors

It is seen in (2) and (3), we introduce the *transformation factors* that are equal to the ratio of the desired maximum weight $W$ to the corresponding perception coefficient (i.e., $c_s$ or $c_a$). In detail, the solution of $(c_s - d^2/c_s)$ in (2) is a set of $[0,c_s]$ and that indicates the impact of distance based on the *Separation* rule perception, whose measurement unit may vary. For example, with $c_s = 5{,}000$ $m$ and maximum weight $W = 30$, the solution set $[0,5000]$ is completely different from the weight range $[0,30]$. Therefore, $T_s$ is used to keep the impact of *Separation* rule unaffected by the measurement unit, which makes our scheme able to adapt to different settings.

To calculate $T_s$, it is required to determine the maximum weight $W$ and the distance $d_{\min}$ when *Separation* will become largely effective for safety issues. Since there is probably the situation where *Migration*, *Alignment*, and *Cohesion* vectors simultaneously point to the same direction, the *Separation* vector needs to point to the opposite direction to prevent any possible collisions. Therefore, the $w_{se}$ should satisfy (7).

$$w_{se} > w_{mi} + w_{al} + w_{co} \tag{7}$$

As inferred from (4) to (6), the maximum values of $w_{mi}$, $w_{al}$, and $w_{co}$ are $2M$, $M$, and $M$, respectively. Consequently, $w_{se}$ should be greater than $4M$ so as to take control of the drone. As a result, we have $(c_s - d_{\min}^2/c_s) \in [0, c_s]$ and $4M \in [0,W]$. Additionally, since $w_{se}$ is proportional to $(c_s - d^2/c_s)$ according to (2), the maximum weight $W$ can be computed by (8).

$$W = \frac{4M}{1 - \dfrac{d_{\min}^2}{c_s^2}} \tag{8}$$

Then, $T_s$ is obtained by (9).

$$T_s = \frac{W}{c_s} = \frac{4M}{c_s - \dfrac{d_{\min}^2}{c_s}} \tag{9}$$

In short, $T_s$ can be computed based on particular preference for the shortest range $d_{\min}$ at which we want the drones begin to exponentially react to *Separation* rule. The act of

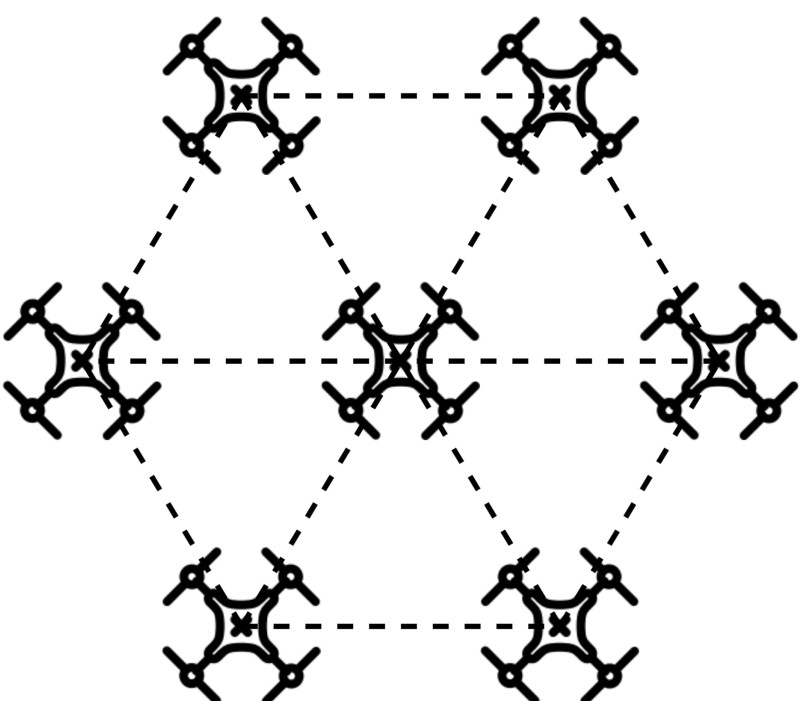

**Figure 1 Ideal formation of a group of quadcopters.**

choosing $d_{\min}$ can be done after $M$ and $c_s$ are determined. For example, if $d_{\min} = 0.4c_s$ then $T_s \approx 4.76M/c_s$. Similarly, the transformation factor $T_a$ for *Avoidance* is determined by (10).

$$T_a = \frac{4M}{c_a - \dfrac{d_{\min}^2}{c_a}} \tag{10}$$

## Rule implementation

The explanations for rules' implementation will be given in prioritized groups. In both *Separation* and *Avoidance*, the drone travels away from nearby objects to avoid crashes, and a repulsion vector for each of them is formed. As we only consider the direction at this stage, these vectors are then normalized. Moreover, an impact called *dist_impact* is also calculated by (11).

$$dist\_impact = c - \frac{d^2}{c} \tag{11}$$

where $c$ is the rule perception that can be $c_s$ or $c_a$ depending on either *Separation* or *Avoidance*, and $d$ is the distance between two drones or between a drone and an obstacle. Subsequently, the normalized vectors are multiplied with *dist_impact* since we are expecting the smaller the distance is, the larger the magnitude of this repellent force is. Then, all resulting vectors are combined and the sum vector is scaled by (12).

$$v_s = \begin{cases} \hat{v} & \text{if } ||v|| > 0 \\ (0,0) & \text{if } ||v|| = 0 \end{cases} \tag{12}$$

---

**Algorithm 1** Separation/Avoidance

**Input:**

*d*: The drone adopting the algorithm

*ob*: List of detected nearby objects, either neighbors or obstacles

*T*: The corresponding transformation factor $T_s$ or $T_a$

**Output:**

*vs*: The scaled vector

1: $v \leftarrow (0,0)$

2: $weight \leftarrow 0$

3: $max\ impact \leftarrow 0$

4: **for** *o* in *ob* **do**

5:     $dist \leftarrow$ distance between *o* and *d*

6:     **if** $dist < c$ **then**

7:         $dist\_impact \leftarrow c - (dist * dist)/c$

8:         $vo \leftarrow o.position - d.position$

9:         $vo \leftarrow dist\_impact * vo/dist$

10:         $v \leftarrow v - vo$

11:         **if** $max\_impact < dist\_impact$ **then**

12:             $max\_impact \leftarrow dist\_impact$

13:         **end if**

14:     **end if**

15: **end for**

16: $vs \leftarrow scale(v)$

17: $weight \leftarrow T * max\_impact$

18: **return** $vs * weight$

---

where $\hat{v}$ is the unit vector of *v*.

In addition, we need to find the nearest object among all detected ones with the most significant impact as *max_impact* inferred from (11) so that the drone will try to avoid the nearest object. Finally, the product of the scaled vector $v_s$, *max_impact*, and *T* produces the return vector. Algorithm 1 describes the implementation of *Separation* and *Avoidance* rules, which are different by the transformation factor *T* (i.e., *T* can be $T_s$ or $T_a$).

The *Migration* rule's effect will attract the drones to the predefined target by forming a vector from the drone to the target point. The vector is then scaled using (12) and assigned a weight affected by the current number of nearby neighbors. If there are any detected neighbors, *Migration* weight will be $M + M/m$ to get individuals in the group to the desired location rather than being stuck by the effects of *Alignment* and *Cohesion*. Otherwise, if there is no detected neighbor, the weight will be *M*. Both cases guarantee that

| Algorithm 2 Migration |
|---|

**Input:**

*d*: The drone adopting the algorithm

*m*: The current number of detected neighbors

**Output:**

*vs*: The scaled vector

1: $v \leftarrow (0,0)$

2: *weight* $\leftarrow 0$

3:    **if** *checkpoint* $\neq$ *null* **then**

4:      $v \leftarrow checkpoint - d.position$

5:      $vs \leftarrow scale(v)$

6:    **if** $m \neq 0$ **then**

7:        $weight \leftarrow (M + M/m)$

8:    **else**

9:        $weight \leftarrow M$

10:   **end if**

11: **else**

12:   $vs \leftarrow (0,0)$

13: **end if**

14: **return** $vs * weight$

*Migration* weight will always be equal or greater than the sum of *Alignment* and *Cohesion* weights. This weighting scheme ensures that even in case both *Alignment* and *Cohesion* velocity vectors point to the same direction but are opposite the target point, the drone will still be able to head towards its target. The implementation of the *Migration* rule is shown in Algorithm 2.

Alignment* and *Cohesion* rules share the same weighting mechanism due to partial similarities in their effect. The implementation of these two rules is represented by Algorithm 3. In detail, the drone will fly in the same direction as its neighbors due to *Alignment*. The vector for this behavior is computed by calculating the average velocity of each drone's nearby neighbors. Regarding *Cohesion*, the drone tries to fly towards the average location of its neighboring drones. The resulting vector is scaled and then multiplied with its weight.

## PERFORMANCE EVALUATION

The proposed weighting mechanism's performance has been evaluated in a flocking control scheme via the simulation developed using Pygame (https://www.pygame.org/wiki/about), a cross-platform and highly portable collection of modules in Python designed to make video games. Although Pygame does not support hardware acceleration and is not powerful in creating 3D graphics, it is well-known for being open-source and

---

**Algorithm 3** Alignment/Cohesion.

**Input:**

*Neighbors*: List of detected neighbors

*m*: The current number of detected neighbors

**Output:**

*vs*: The scaled vector

1: $v \leftarrow (0,0)$
2: *weight* $\leftarrow 0$
3: **if** $m \neq 0$ **then**
4:     **if** Alignment **then**
5:         **for** each $n$ in *Neighbors* **do**
6:             $v \leftarrow v + n.velocity$
7:         **end for**
8:     **else**
9:         **for** each $n$ in *Neighbors* **do**
10:             $v \leftarrow v + n.position$
11:         **end for**
12:     **end if**
13:     $v \leftarrow v/m$
14:     $vs \leftarrow scale(v)$
15:     *weight* $\leftarrow (M/m)$
16: **else**
17:     $vs \leftarrow (0,0)$
18: **end if**
19: **return** $vs$ * *weights*

---

lightweight compared to other tools. Therefore, Pygame is an appropriate choice considering the scope of our implementation.

## Evaluation metrics

Regarding the evaluation process, the proposed scheme is compared to the conventional one using the weighted sum of all rule vectors to assess their performance differences in terms of two metrics, including flock compactness and collision reduction represented by the number of crashed drones.

- **Flock compactness** shows the drones' capability in maintaining the optimal distances among one another by calculating the average distance of each pair of drones in the flock. The smaller the average distance is, the less space the flock uses. This is particularly useful for flights in confined environments or using wireless communication. This metric is represented by the average distance computed by (13).

$$\bar{d} = \frac{1}{N_s} \sum_{k=1}^{N_s} \frac{2}{n(n-1)} \sum_{i=1}^{n-1} \sum_{j=i+1}^{n} d_{i,j} \qquad (13)$$

where $N_s$ is the number of loops in the swarm algorithm, $n$ is the number of drones, and $d_{i,j}$ is the distance between drone $i$ and $j$.

- **The number of crashed drones** is counted if there are any other objects (i.e., obstacles or neighboring drones) within the collision distance around a drone.

Besides, the magnitude of each rule's velocity vector before being combined into one final vector will be investigated over how each swarm behavior influences the drones' movement trajectory during flights.

## Methodology

In the simulation program, flight tests are conducted within 1,200 x 700 on a 2D plane measure by **unit**. The unit can be pixels or scalable to different measures since the proposed transformation factors keep the rules weighting computation unaffected by the measurement unit, as presented above.

For both conventional and proposed schemes, all common parameters that represent each drone's characteristics are summarized in Table 1. Drones are assumed to fit in a circle with a diameter of 8 units, and the center coordinates are used as their position. Besides, a collision is determined once any objects within each drone's safety zone are defined by a collision distance of 8 units (same as drone's size). In addition, *Alignment* and *Cohesion* behaviors start taking effect when there are neighbors closer than their perceptions (50 units). Meanwhile, *Separation* and *Avoidance* will only become operative with a smaller perception (40 units) to prevent the unnecessary impact that may cause drones to be directed away from their group early. The speed of each drone will be scaled to 1.2 units/step if it exceeds this value.

To evaluate our proposed scheme's effectiveness compared with the conventional one adopting fixed weights, it is required to determine an optimal set of fixed rule weights that produce the conventional scheme's best experimental results. Therefore, we conduct several simulations with *Separation* and *Avoidance*'s weight being gradually increased, whereas other rules' weights are fixed. The reason is that with higher priorities, *Separation* and *Avoidance* rules have the most considerable influence on both the flock's compactness and the number of crashed drones.

It is reasonable to start with the weight set of $\{w_{se}:w_{av}:w_{mi}:w_{al}:w_{co}\}$ as $\{4:4:2:1:1\}$ based on the proposed rule prioritization and the rule's weights computed by (2), (3), and (7). For every weight set, 100 simulations are executed with initial randomized positions of drones and obstacles. As can be seen in Fig. 2, the drones being too close leads to the large number of crashed drones. Thus, *Separation* and *Avoidance* weights are increased by 0.4 in each next experiment. It is seen that when the *Separation* and *Avoidance* weights begin to reach 5.2 and higher, the number of crashed drones remains steady at nearly 10. However, the flock compactness tends to rise along with the increase in weights of *Separation* and *Avoidance*. Regarding safety concerns and the flock compactness,

**Table 1 Swarm members' characteristic parameters.**

| Parameter | Value |
|---|---|
| Drone size | 8 units |
| Collision distance | 8 units |
| Obstacle | 35 units × 35 units |
| *Alignment* perception | 50 units |
| *Cohesion* perception | 50 units |
| *Separation* perception | 40 units |
| *Avoidance* perception | 40 units |
| Maximum speed | 1.2 units/step |

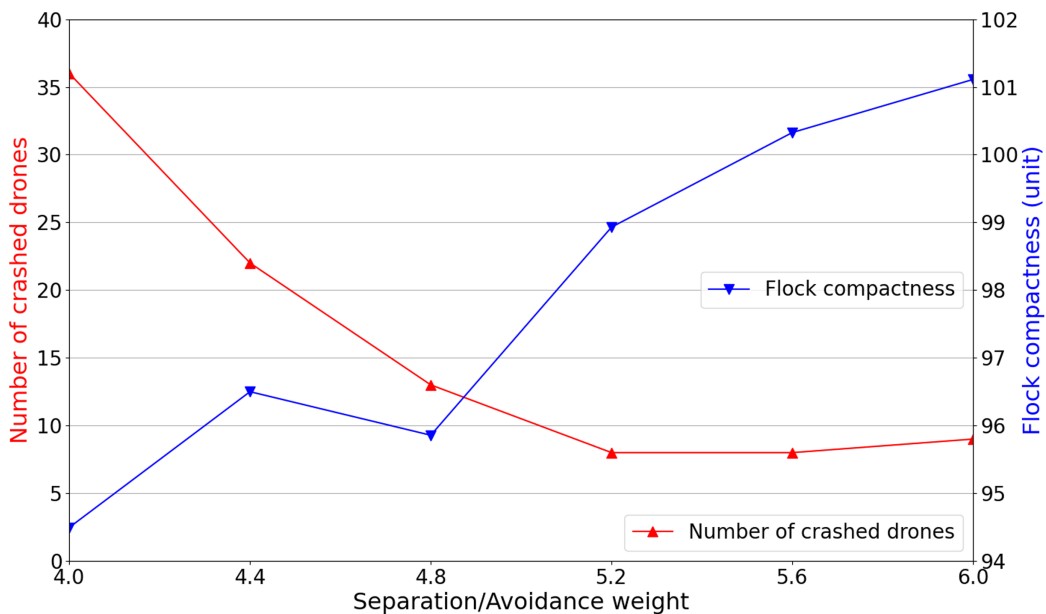

**Figure 2** *Separation* and *avoidance* weights' variation and their corresponding simulation results.

*Separation* and *Avoidance* weights will be set as 5.2 (i.e., $w_{se} = w_{av} = 5.2$) for further performance comparison with our proposed scheme.

The optimal weight set of {5.2:5.2:2:1:1}, along with our adaptive weighting mechanism, are ready for further comparisons. For each simulation, ten drones are initially created with random positions near the upper left corner of the window, and then they proceed to move towards the target point at the lower right corner. There are two scenarios, as follows:

- **Flights without obstacles:** this is the ideal environment for the deployment of any flocking control scheme. As there is no *Avoidance* behavior, the *Separation* velocity vectors keep being dominant during the entire flights, leading to no collision between drones. Therefore, flock compactness is the only metric for evaluation and comparison in this scenario.

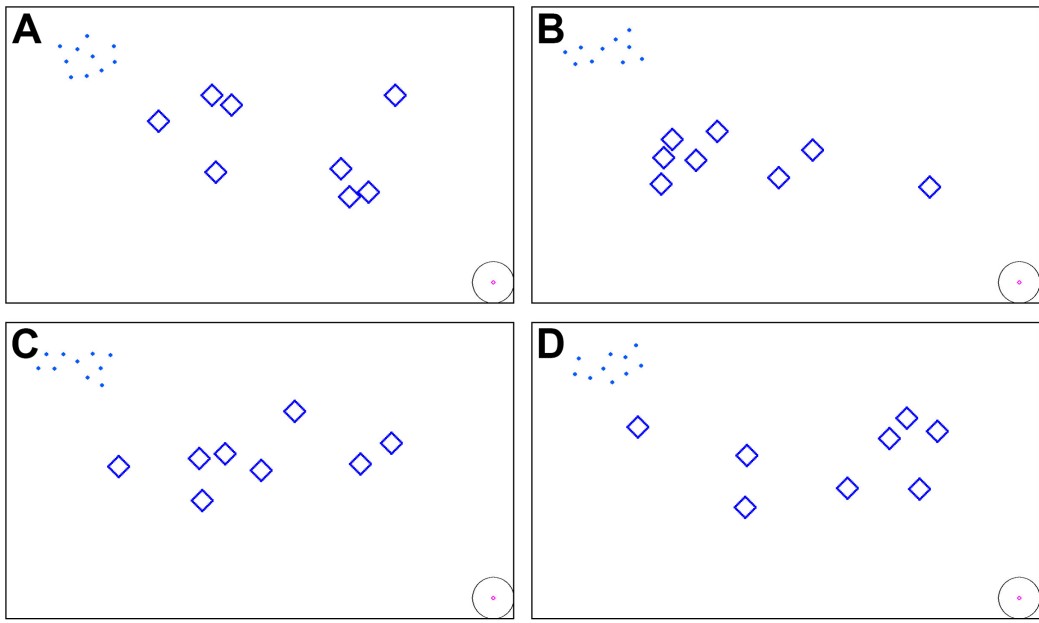

**Figure 3** (A–D) Initial positions of the swarm members and obstacle shapes being randomized for each simulation.

- **Flights with obstacles:** obstacles are randomly generated in an area of $800 \times 300$ in the middle of the window for each simulation, as depicted in Fig. 3. For this scenario, all two metrics are used.

## Simulation results

An ensemble of 200 simulations has been done for each scheme in all scenarios. Fig. 4 summarizes the simulation results regarding the flock compactness in the obstacle-free scenario. It can be noted that the drones have an average distance between each other of about 81 units with the conventional flocking control scheme. Meanwhile, our proposed approach produces approximately 73 units. This indicates that by leveraging our improved scheme, the flock is better at maintaining its compactness with roughly 10% smaller average distance.

Figure 5 shows the average distance of each scheme when it comes to flying areas with obstacles. It can be noticed that in terms of the flock compactness, the proposed scheme continues to outperform its conventional counterpart. In fact, every pair of swarm members in the proposed scheme manages to maintain an average distance of 85 units, which is 14.14% lower than the conventional scheme's result of 99 units.

Besides maintaining a smaller relative distance with other neighbors, the drones maneuvered by the proposed scheme happen to encounter fewer crashes. In detail, the conflict of *Separation* and *Avoidance* rules in the conventional scheme leads to 15 crashed drones during 200 flight tests. Meanwhile, due to the flexibility in weight allocation, the proposed scheme keeps the flock free from collisions.

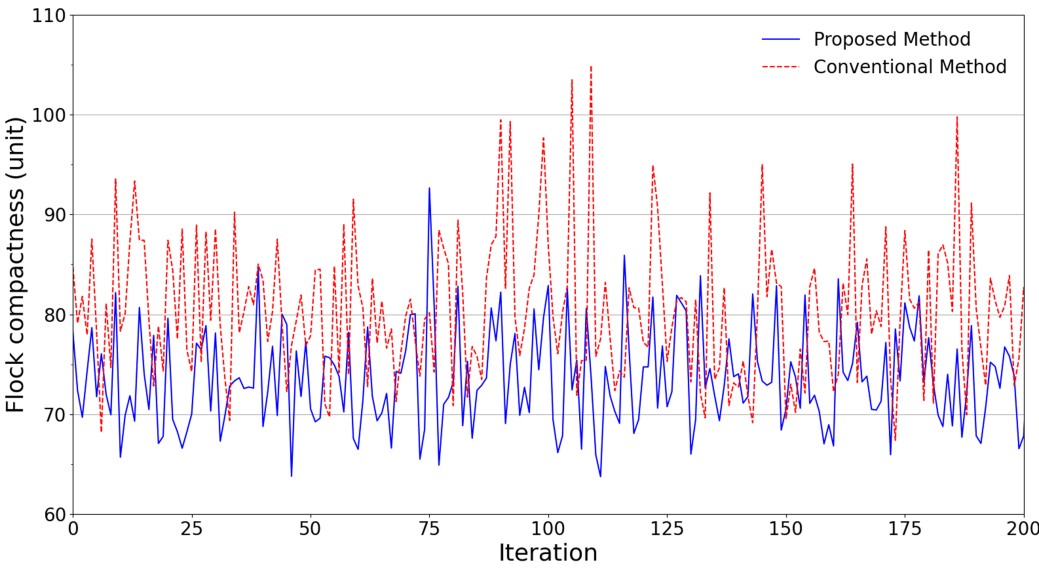

**Figure 4 Experimental results in terms of flock compactness obtained from scenarios without obstacles.**

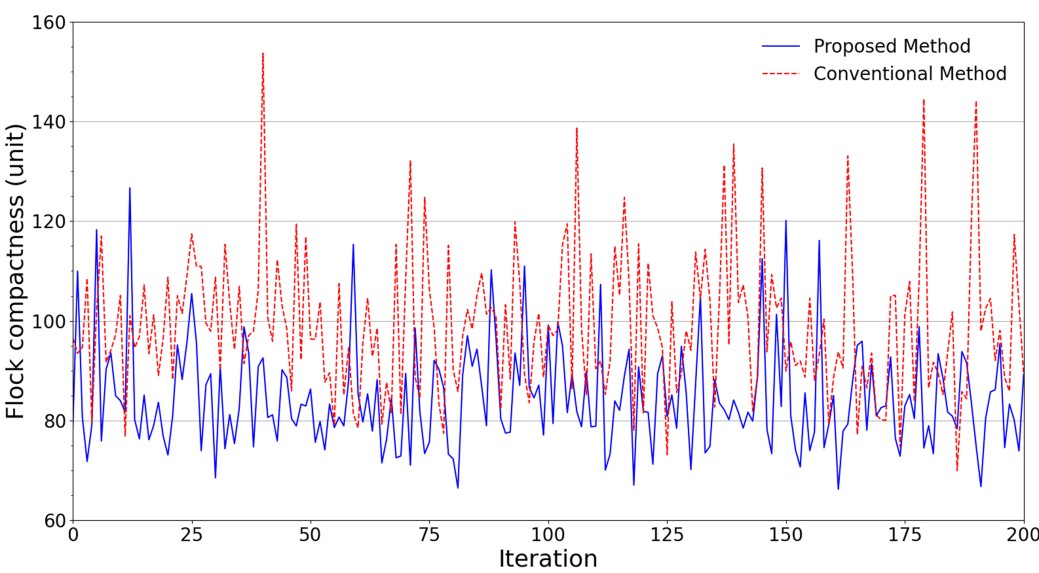

**Figure 5 Experimental results in terms of flock compactness obtained from scenarios with obstacles.**

To explain the reason behind this better performance of the proposed scheme, we investigate every swarm rule's impact on a drone's behavior against the conventional control scheme. As shown in Fig. 6, it is observed that there are numerous times where the sudden detection of both nearby neighbors and obstacles leads to unexpected behaviors of the drone. For instance, at around simulation step 300, the sudden and considerably weighted appearance of *Avoidance* rule in purple leads the drone to be directed far from its neighbors. This is to the extent that *Separation* force is barely needed and *Cohesion* and *Alignment* forces are predominant to keep the drone closer. In contrast, regarding the

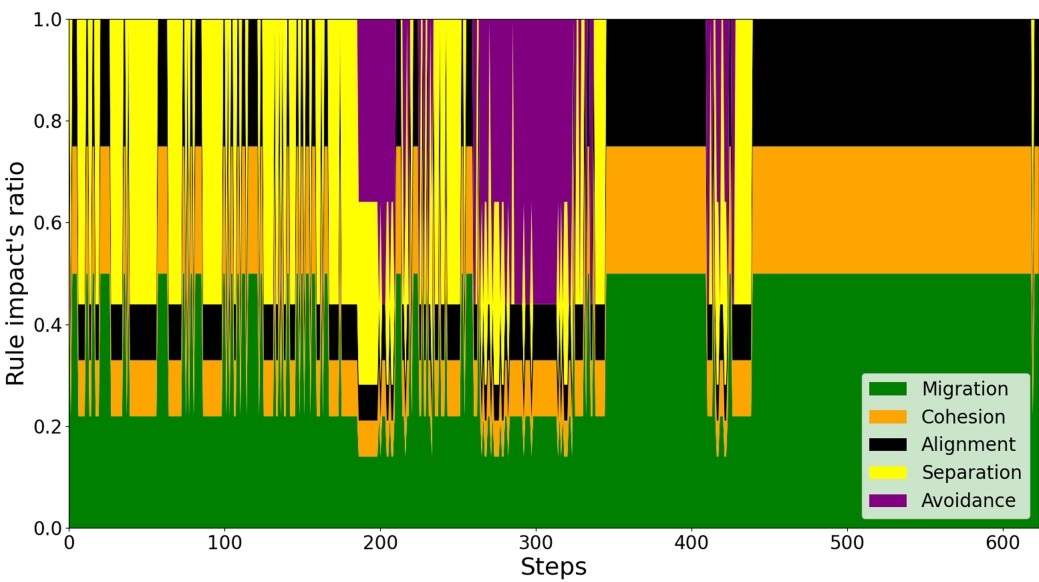

**Figure 6 The effects of swarm rules on a drone's behaviors by the conventional method.**

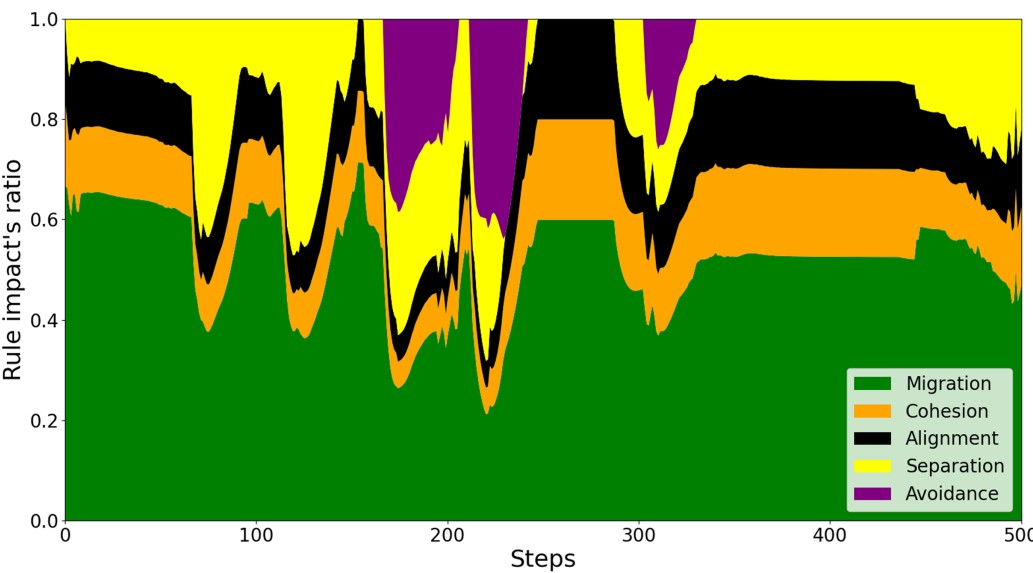

**Figure 7 The effects of swarm rules on a drone's behaviors by the proposed method.**

proposed model, as investigated in Fig. 7, the adaptive weighting mechanism enables the drone to stay close in the swarm throughout the flight. This is indicated by all of the swarm behaviors' presence during and after avoiding the obstacles (around steps 200, 300, and 410).

## CONCLUSIONS

We propose an adaptive weighting mechanism to compute each behavioral velocity vector for a flocking control scheme adopting conventional Reynolds rules. The simulation results

show that our proposed scheme has better performance than the conventional one in terms of flock compactness (smaller up to 14.14%), collision reduction (e.g., free from collisions under various scenarios), and how the swarm rules are retained during flights. In our proposed weighting mechanism, the transformation factors determined by (9) and (10) can keep the impact of the swarm rules unaffected by the measurement unit, which makes our scheme able to adapt to different settings. Therefore, our simulation results are reliable.

However, the current simulations are performed in an ideal environment where the flights occur without impact from external forces. For instance, in the real world, strong winds may affect the quadcopter flight stability. Combining our flocking control scheme with calculations of such forces will further prove our approach's practical applicability. Moreover, our future study will develop more simulation scenarios in which the drones fly at different altitudes before deploying the proposed control scheme on physical drones for practical performance evaluation.

### Funding
This research is funded by Ho Chi Minh City University of Technology (HCMUT), VNU-HCM under grant number HCMUT-002603-2021. The funders had no role in study design, data collection and analysis, decision to publish, or preparation of the manuscript.

### Grant Disclosures
The following grant information was disclosed by the authors:
Ho Chi Minh City University of Technology (HCMUT), VNU-HCM: HCMUT-002603-2021.

### Competing Interests
The authors declare that they have no competing interests.

### Author Contributions
- Duc N. M. Hoang conceived and designed the experiments, performed the experiments, analyzed the data, performed the computation work, prepared figures and/or tables, and approved the final draft.
- Duc M. Tran conceived and designed the experiments, performed the experiments, analyzed the data, performed the computation work, prepared figures and/or tables, and approved the final draft.
- Thanh-Sang Tran conceived and designed the experiments, performed the experiments, performed the computation work, prepared figures and/or tables, and approved the final draft.
- Hoang-Anh Pham conceived and designed the experiments, analyzed the data, authored or reviewed drafts of the paper, and approved the final draft.

## Data Availability

The data and code are available at GitHub: https://github.com/minhduccse/adaptive-weighting-mechanism.

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
