# Peer review of "An adaptive weighting mechanism for Reynolds rules-based flocking control scheme"

_PeerJ Computer Science, doi:10.7717/peerj-cs.388_

## Round 0.1 · original submission · Minor Revisions

This paper presents a novel control scheme for a swarm of UAVs, this paper should be revised aligned with the reviewer's comments.

Reviewer 1 ·

Basic reporting

no comment

Experimental design

no comment

Validity of the findings

no comment

Additional comments

This paper proposes a novel control scheme for a swarm of UAVs that
employs the original Reynolds rules but adopts an adaptive weight allocation
mechanism based on the current context than being fixed at the beginning.

This paper contains some interesting results. Please find the following comments that may help improve the paper.

(1) It is suggested to highlight the challenges of the investigated problem.

(2) The significance of using an adaptive weight allocation mechanism needs to be explained.

(3) What is the meaning of step 3 in algorithm 1?

(4) Please explain the steps 6-9 in algorithm 2.

(5) The simulation is done on Pygame. What is the advantage and disadvantage compared to other software?

(6) This paper is about flocking control. The following three references should be included in the introduction section to present readers a complete literature review. B. Shucker, T. D. Murphey, and J. K. Bennett, “Convergence-preserving switching for topology-dependent decentralized systems,” IEEE Trans. Robot., vol. 24, no. 6, pp. 1405-1415, Dec. 2008. B. Ning, Q.-L. Han, Z. Zuo, J. Jin, and J. Zheng, “Collective behaviors of mobile robots beyond the nearest neighbor rules with switching topology,” IEEE Trans. Cybern., vol. 48, no. 5, pp. 1577-1590, May 2018. R. Olfati-Saber, “Flocking for multi-agent dynamic systems: Algorithms and theory,” IEEE Trans. Autom. Control, vol. 51, no. 3, pp. 401-420, Mar. 2006.

(7) In figure 2, the decrease of average distance from 4.4 to 4.8 looks strange, please explain the reason.

(8) The language of this paper needs to be improved.

Reviewer 2 ·

Basic reporting

The paper has been written in reasonably good English. Introduction and background may be enough for the specialist. In general, it would be nice to have a bit more details. There are over 20 relatively recent references. Also, major work from the past is pointed out. Figures are generally well together with the text. Perhaps a more rigorous usage of the units on axis, whenever applicable, would be relevant. For example, distance should have units, even if it is just “unit” (explained in the text).

Experimental design

The experiment is purely software-based, which may be enough for the current paper, but some real-life validation would enhance the value. So far the impact of factors (external forces) making it closer to the real-life is proposed only in upcoming works.

Validity of the findings

Better comparison with state of the art would be in order. No doubt that the results are interesting and useful, should the findings be verified in real life and against competing solutions.

Additional comments

It may be that the notation used in line68 is acceptable in the specialist circles (if time is taken as unity for example), but generally adding space coordinates in m, “unit” or similar (position vector) to the speed (velocity vector) in m/s, “unit”/s or similar should be avoided. There seems to be some problem with indexes - ws (line 168) seems to be alone.

---

## Round 0.2 · Minor Revisions

The math equations must be explained in how will be used in your study. No further comments

---

## Round 0.3 · accepted · Accept

The author has met all the comments provided by all reviewers and the the paper is in very good standard for publishing in PeerJ computer science journal.